# A route to metalloligands consolidated silver nanoclusters by grafting thiacalix[4]arene onto polyoxovanadates

Zhi Wang [1,2], Yan-Jie Zhu[1,2], Bao-Liang Han[1], Yi-Zhi Li[1], Chen-Ho Tung[1] & Di Sun [1] ✉

Metalloligands provide a potent strategy for manipulating the surface metal arrangements of metal nanoclusters, but their synthesis and subsequent installation onto metal nanoclusters remains a significant challenge. Herein, two atomically precise silver nanoclusters {$Ag_{14}[(TC4A)_6(V_9O_{16})](CyS)_3$} (Ag14) and {$Ag_{43}S[(TC4A)_2(V_4O_9)]_3(CyS)_9(PhCOO)_3Cl_3(SO_4)_4(DMF)_3 \cdot 6DMF$} (Ag43) are synthesized by controlling reaction temperature ($H_4TC4A = p$-tert-butyl-thiacalix[4]arene). Interestingly, the 3D scaffold-like $[(TC4A)_6(V_9O_{16})]^{11-}$ metalloligand in Ag14 and 1D arcuate $[(TC4A)_2(V_4O_9)]^{6-}$ metalloligand in Ag43 exhibit a dual role that is the internal polyoxovanadates as anion template and the surface $TC4A^{4-}$ as the passivating agent. Furthermore, the thermal-induced structure transformation between Ag14 and Ag43 is achieved based on the temperature-dependent assembly process. Ag14 shows superior photothermal conversion performance than Ag43 in solid state indicating its potential for remote laser ignition. Here, we show the potential of two thiacalix[4]arene modified polyoxovanadates metalloligands in the assembly of metal nanoclusters and provide a cornerstone for the remote laser ignition applications of silver nanoclusters.

Silver nanoclusters (NCs) have been gaining attention due to their various applications in fields such as photoluminescence, catalysis, optical imaging, and biology[1–4]. In recent years, the synthetic chemistry of silver NCs has advanced significantly through ligand engineering and template strategies, resulting in highly controllable and precise synthesis[5,6]. The protective ligand is critical in determining the structure, stability, and properties of silver NCs. Organic ligands such as thiols, alkynes, and phosphines have been commonly used, and recent research has expanded to nitrogen-donor ligands, metalloligands, and macrocyclic hosts[7–16]. Among these, metalloligands have brought opportunities for shaping ordered architectures and enhancing the stability of silver NCs. Zhang group reported a series of stable Ag–Ti NCs protected by a flexible trifurcate $TiL_3$ metalloligand, in which the $TiL_3$ moieties serve as the capping vertices of the silver NCs, thereby forming the tetrahedral geometry[11]. Recently, our group successfully achieved several $Mo^{VI}$-anchored thiacalix[4]arene metalloligands-protected silver NCs with different geometries through a stepwise assembly strategy, wherein it was demonstrated that the solvent controlled the cyclization of the metalloligand and then modulated the final silver NCs[12]. It is apparent from the aforementioned results that the thiacalix[4]arene as a derivative of calixarene has phenolic hydroxyl groups and bridging sulfide groups, which again finds a niche for heterobimetallic cluster assembly[17,18].

Compared with transition metals (e.g., Cr, Mo, and W), vanadium is a multivalent redox-sensitive element that can exist in oxidation states ranging from −1 to +5, with the three highest oxidation states, +3, +4, and +5, being most common in the natural environment[19]. Poly-oxovanadates (POVs) have unique characteristics that set them apart from polyoxometallates (POMs). They exhibit rich redox behavior, intriguing structural topologies, and flexible coordination patterns

[1]School of Chemistry and Chemical Engineering, State Key Laboratory of Crystal Materials, Shandong University, Ji'nan 250100, People's Republic of China. [2]These authors contributed equally: Zhi Wang, Yan-Jie Zhu. ✉e-mail: dsun@sdu.edu.cn

with secondary metal ions[20,21], making them excellent anion templates in the synthesis of silver NCs such as $[(V_{10}O_{28})@Ag_{50}]$, $[(V_{10}O_{28})@Ag_{44}]_n$ and $[(V_{10}O_{28})@Ag_{46}]_n$[22,23]. In addition to being an anionic template, the inorganic POVs can also be grafted by organic functional groups to form the hybrid POVs that will further enrich the structural chemistry of silver NCs. Xie and coworkers synthesized a series of Ag–V hybrid NCs where phosphonate-modified oxovanadate building blocks or POVs were revealed on the surface of the metal shell[24–26]. Herein, we envisioned that integrating thiacalix[4]arenes and POVs into an entity can provide the dual function of multi-dentate chelating of thiacalix[4]arenes and anionic templating of POVs, leading to the emergence of richer coordination patterns and assembly phenomena than using organic ligands.

In this work, two silver NCs of $\{Ag_{14}[(TC4A)_6(V_9O_{16})](CyS)_3\}$ (**Ag14**) and $\{Ag_{43}S[(TC4A)_2(V_4O_9)]_3(CyS)_9(PhCOO)_3Cl_3(SO_4)_4(DMF)_3\cdot6DMF\}$ (**Ag43**) consolidated by 3D and 1D TC4A⁴⁻-POVs metalloligands are isolated by adjusting the solvothermal reaction temperature under otherwise identical conditions and characterized by single crystal X-ray diffraction (SCXRD) ($H_4TC4A = p$-tert-butylthiacalix[4]arene). The two TC4A⁴⁻-POVs metalloligands successfully integrate the dual role of classical metalloligand and anion template, which were previously unobserved in the realm of both silver NCs and POMs chemistry. The structural conversion from **Ag43** to **Ag14** can be achieved by only adjusting temperature, whereas the reverse structural conversion needs stimuli from both temperature adjustment and reactant additives. Moreover, **Ag14** possesses superior photothermal conversion performance, which shows promising applications in laser ignition materials and photothermal therapy.

## Results

### Synthesis discussion

The mixture of $H_4TC4A$, $VOSO_4\cdot xH_2O$, and PhCOOAg was dispersed in DMF (N, N-dimethylformamide) and underwent solvothermal reaction to evaluate the feasibility of the TC4A⁴⁻-POVs metalloligands in the assembly of silver NCs. However, only a simple metal complex $\{Ag_2(TC4A\text{-}VO)_2(DMF)_2\cdot6DMF\}$ (**Ag2**) was obtained under the above system, which indicates the possibility of the metalloligand as an ideal candidate for heterobimetallic assembly (Supplementary Fig. 1). Moreover, previously reported oxovanadium $H_4TC4A$ complexes such as $PPh_4[(H_2TC4A)VOCl_2]$, $(PPh_4)_2[\{(H_2TC4A)V(O)(\mu\text{-}O)\}_2]$ and $PPh_4[(TC4A)V=O]$, etc. also suggested that the vanadium ion can ligate with $H_4TC4A$ through phenolic hydroxyl oxygens and thioether groups (Supplementary Fig. 2)[27], which fully illustrated the diversity of the coordination modes of $H_4TC4A$ with vanadium ions.

Motivated by the structure of **Ag2**, the auxiliary ligand of thiolate was introduced into the subsequent synthesis reactions. By varying the solvothermal reaction temperature, the pure phase of **Ag14** and a mixed phase of **Ag14** and **Ag43** were obtained at 65 and 75 °C, respectively. Interestingly, we found 3D scaffold-like $[(TC4A)_6(V_9O_{16})]^{11-}$ metalloligand in **Ag14** and 1D arcuate $[(TC4A)_2(V_4O_9)]^{6-}$ metalloligand in **Ag43** (Fig. 1). To investigate the effect of temperature on the assembly, solvothermal reactions were performed in the range of 65–120 °C (Supplementary Fig. 3). Interestingly, **Ag14** can form in the wide temperature range of 65–120 °C whereas the mixture of **Ag14** and **Ag43** was found in the narrow temperature range of 75–80 °C. In spite of many attempts, the pure phase of **Ag43** cannot be obtained by simple temperature manipulation. Benefiting from the obviously different shapes, the clump-like **Ag14** and block-like **Ag43**, we can readily separate them manually under the microscope. Apparently, temperature has significant effects on the coordination-driven assembly process of two silver NCs, providing possibilities for thermal-induced structure transformation. Combining the reaction temperature as well as the compositions of the two silver NCs, the structure transformation of two silver NCs was successfully achieved by thermal stimulus. In addition, the vanadium source in the synthesis of **Ag14** is non-specific, as evidenced by the successful crystallization of **Ag14** after solvothermal reaction using other vanadium regents, such as $NaVO_3$ or $Na_3VO_4$ under otherwise identical conditions. The crystallography-related information and other characterizations, including infrared (IR) spectroscopy, ultraviolet–visible (UV–Vis) spectroscopy, and energy dispersive spectroscopy (EDS), are collected in Supplementary Figs. 4–32 and Supplementary Tables 1 and 2.

### X-ray crystal structures

**Ag14** and **Ag43** crystallized in the triclinic $P−1$ and trigonal $R−3$ space groups, and the asymmetric units of two NCs contain the complete cluster and 1/3 of the complete cluster, respectively. As shown in Fig. 2a, **Ag14** contains 14 silver ions, one 3D scaffold-like $[(TC4A)_6(V_9O_{16})]^{11-}$ metalloligand and three CyS⁻. The configuration of the inner POVs and the binding feature of the interfacial macrocyclic TC4A⁴⁻ are intriguing in 3D $[(TC4A)_6(V_9O_{16})]^{11-}$ metalloligand. In detail, six V atoms are attached to the lower rim of TC4A⁴⁻ via V–O$_{phenol}$ bonds and adopt an octahedral coordination pattern with two other oxygen atoms, one stemmed from the $\{VO_4\}$ tetrahedron and the other from a terminal oxygen atom, to form six TC4A-VO₂ units. These TC4A-VO₂ units are connected to the six oxygen atoms via a vertex-sharing pattern on two poles of the rod-like $[V_3O_{10}]^{5-}$ anion that consists of three vertex-sharing $\{VO_4\}$ tetrahedra, resulting in the formation of $[(TC4A)_6(V_9O_{16})]^{11-}$ (Fig. 2b). Here, each of the upper and lower poles of $[V_9O_{16}]^{11-}$ has three triangularly distributed $\{VO_6\}$ connected by rod-like $[V_3O_{10}]^{5-}$ by sharing vertices (Supplementary Fig. 4a). The V–O bond lengths of $\{VO_4\}$ and $\{VO_6\}$ lie in the range of 1.583–1.870 Å and 1.579–2.108 Å, respectively. From the top view, the two groups of $\{VO_6\}$ in the triangular geometry are not face-to-face but rotated about 18.8°

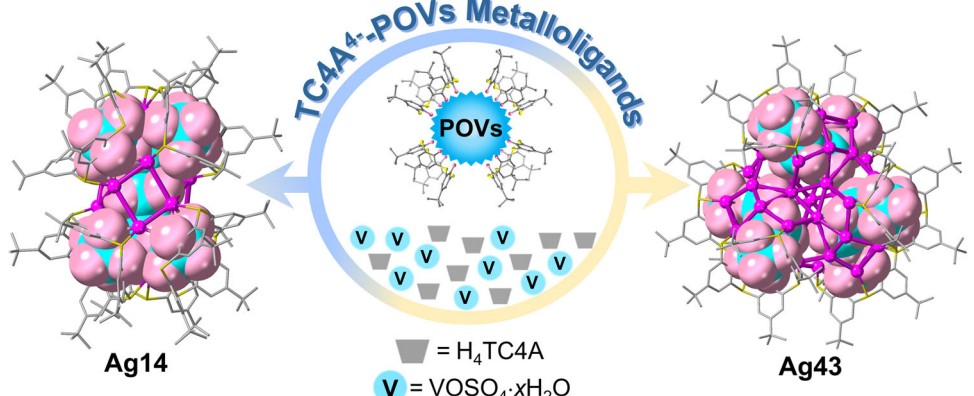

**Fig. 1 | Description of TC4A⁴⁻-POVs metalloligands.** Schematic diagram of the assembly of TC4A⁴⁻-POVs metalloligands, $H_4TC4A = p$-tert-butylthiacalix[4]arene, POVs = polyoxovanadates. Color labels: purple, Ag; yellow, S; gray, C; pink, O; cyan, V.

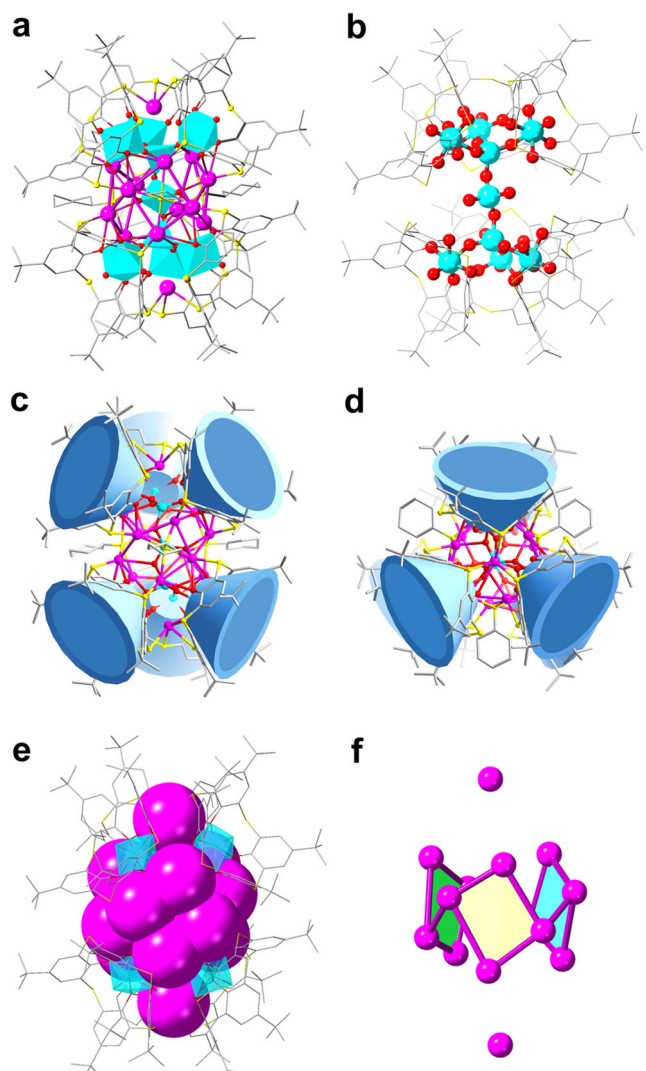

**Fig. 2 | Molecular structure of the Ag14 resolved by SCXRD. a** Total structure of the **Ag14**; **b** the structure of 3D scaffold-like $[(TC4A)_6(V_9O_{16})]^{11-}$ metalloligand; The side view (**c**) and the top view (**d**) of the distribution of $TC4A^{4-}$ in **Ag14**; **e** the $Ag_{14}$ shell cemented by a 3D scaffold-like $[(TC4A)_6(V_9O_{16})]^{11-}$ metalloligand; **f** the $Ag_{14}$ silver shell. Color labels: purple, Ag; yellow, S; gray, C; red, O; cyan, V; cyan polyhedron, POVs; blue cup, $TC4A^{4-}$.

with respect to each other (Supplementary Fig. 4b), which causes the distortion of the outer silver shell and reduces the overall symmetry of the cluster. The coordination numbers of silver atoms in **Ag14** are three (1 Ag atom in $AgS_2O$, and 2 in $AgS_3$), four (4 in $AgS_2O_2$), and five (7 in $AgS_2O_3$).

Each group of three $TC4A^{4-}$ is located at the upper and lower poles of **Ag14** (Fig. 2c, d). Notably, the 3D $[(TC4A)_6(V_9O_{16})]^{11-}$ metalloligand runs through the silver shell and ligates all the silver atoms through Ag–S and Ag–O bonds (Supplementary Fig. 5). The waist of $[(TC4A)_6(V_9O_{16})]^{11-}$ is surrounded by 12 silver atoms that resemble a silver crown consisting of three parallelograms connected through Ag···Ag interactions. The remaining two silver atoms suspended at the upper and lower parts of the silver crown are fixed in the center of the 3D scaffold by Ag–S bonds (Figs. 2e, f). Besides the 3D $[(TC4A)_6(V_9O_{16})]^{11-}$ metalloligand, the 3 $CyS^-$ exhibit $\mu_4$ bonding mode capped on the $Ag_{12}$ crown with the Ag–S bond lengths ranging from 2.440 to 2.718 Å (Supplementary Fig. 6).

As clarified in Fig. 3a, **Ag43** is composed of an $Ag_{43}$ shell, an $S^{2-}$ anion, three 1D arcuate $[(TC4A)_2(V_4O_9)]^{6-}$ metalloligands, the organic

and inorganic mixed ligand shell of 9 $CyS^-$, 3 $PhCOO^-$, 4 $SO_4^{2-}$, 3 $Cl^-$, and three coordinated DMF molecules. The neutral **Ag43** has $C_3$ symmetry with the crystallographic $C_3$ axis passing through the Ag15, $S^{2-}$ and $SO_4^{2-}$ (Supplementary Fig. 7). No sulfur-releasing reagent was added to the reaction, so the $S^{2-}$ was probably derived from the $(CySAg)_n$ precursor[28]. **Ag43** has intriguing interfacial binding profiles resulting from the synergistic coordination of 1D arcuate metalloligands and other auxiliary ligands. The structure of the 1D arcuate metalloligand is disparate to that found in **Ag14**, where two $TC4A-VO_2$ units are connected by a $\{V_2O_7\}$ (V–O: 1.646–1.820 Å) by sharing vertices (Supplementary Fig. 8). Specifically, the structure of **Ag43** can be described as three 1D arcuate metalloligands ligating 15 silver atoms to form $Ag_{15}$ caps, which are fused by sharing silver vertices (black atoms) to form an $Ag_{42}$ metallic skeleton that further traps an Ag15 (purple atom) atom situated on the crystallographic $C_3$ axis to build the final $Ag_{43}$ shell (Ag···Ag distances: 2.910–3.242 Å) (Fig. 3b). The interior of the $Ag_{43}$ shell is further reinforced by an $S^{2-}$ anion in a $\mu_7$ coordination pattern and the surface coordination vacancies are filled with other auxiliary ligands. In detail, the silver shell of **Ag43** is shamrock-shaped, with the cavity large enough to accommodate three $\{V_4O_9\}$ acting as anion templates (Figs. 3c, d), and six $TC4A^{4-}$ are combined in pairs by $\{V_4O_9\}$ and divided into three groups evenly arranged around $Ag_{43}$ shell (Figs. 3e, f). Three $PhCOO^-$ adopt the unified $\mu_3-\kappa^2:\kappa^1$ coordination mode toward silver atoms to fill in the interstice between two $TC4A^{4-}$ in the 1D arcuate metalloligand (Ag–O distances: 2.332–2.445 Å) (Supplementary Fig. 9), realizing the reinforcement of the periphery of the $Ag_{43}$ shell. Nine $CyS^-$ ligands surround the periphery of the $Ag_{43}$ shell in $\mu_4$ coordination mode, with six of them locating around the $PhCOO^-$ ligands and the remaining three capping on the interspace between three 1D arcuate metalloligands (Supplementary Fig. 10). In addition, the region of the $Ag_{43}$ shell near to the $C_3$ axis is further cemented by inorganic anions of $SO_4^{2-}$ and $Cl^-$. The $SO_4^{2-}$ anion passing through the crystallographic $C_3$ axis adopts $\mu_6-\kappa^3:\kappa^1:\kappa^1:\kappa^1$ coordination mode toward Ag (Ag–O distances: 2.270–2.81 Å) and the other three $SO_4^{2-}$ anions arranged in a triangular pattern adopt a unified $\mu_6-\kappa^2:\kappa^2:\kappa^2$ mode (Ag–O distances: 2.264–2.684 Å) (Supplementary Fig. 11a). Each $Cl^-$ anion on the surface of the silver shell is coordinate to five Ag atoms, acting as inorganic ligands and delivering complementary surface binding due to their small size (Supplementary Fig. 11b). Additionally, three DMF molecules are also involved in the stabilization of the silver shell. **Ag43** is protected by metalloligands, organic and inorganic mixed ligand shell, with the silver atoms forming the coordination numbers of three (three Ag atoms in $AgS_2O$, and three in $AgSO_2$), four (three in $AgS_2O_2$, three in $AgSO_2Cl$, and three in $AgS_2OCl$), and five (nine in $AgS_2O_3$, six in $AgSO_3Cl$, three in $AgS_2O_2Cl$, one in $AgSO_3Cl$, six in $AgSO_4$, and three in $AgS_2O_3$).

Based on the above structural analysis, the reasons for **Ag14** can be obtained over a wide temperature range of 65–120 °C are summarized as follows: (i) The crystal structure of **Ag14** is simple, consisting of an $Ag_{14}$ shell and binary ligand combination of $[(TC4A)_6(V_9O_{16})]^{11-}$ metalloligand and $CyS^-$; and (ii) The 3D scaffold-like $[(TC4A)_6(V_9O_{16})]^{11-}$ metalloligand penetrating the cluster counteracts the local positive charge and increases the structural stability. Most silver NCs can only be isolated at a specific temperature or within a narrow temperature range, which requires high-precision temperature control equipment and synthesis conditions. On the other hand, silver NCs that can be obtained over a wide temperature range offer great advantages in their synthesis and applications. Furthermore, the most attractive aspects of **Ag14** and **Ag43** are the 3D scaffold-like $[(TC4A)_6(V_9O_{16})]^{11-}$ metalloligand and 1D arcuate $[(TC4A)_2(V_4O_9)]^{6-}$ metalloligand, respectively, which exhibit a dual role in the assembly process of two silver NCs: (i) inner POVs part exerts the anion template effect; and (ii) the $TC4A^{4-}$ on the surface as a passivator to stabilize the whole NC. In addition, all of the vanadium cations in both clusters are in their highest oxidation state of +5, which has a smaller ion radius as

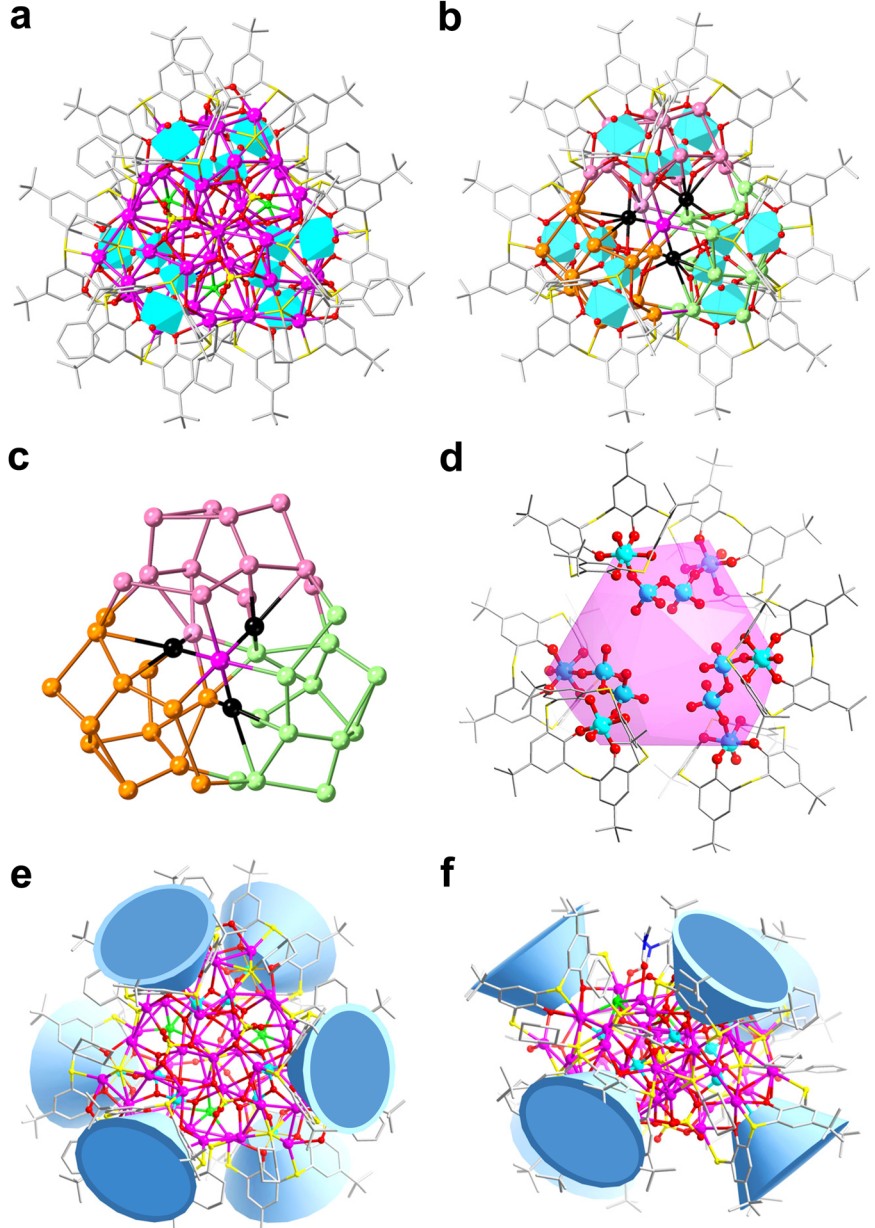

**Fig. 3 | Molecular structure of the Ag43 resolved by SCXRD. a** Total structure of the **Ag43; b** the $Ag_{43}$ shell cemented by three 1D arcuate metalloligands; **c** the $Ag_{43}$ shell in (**b**), each $Ag_{15}$ caps highlighted individually by different colors (pink, orange and green), silver atoms shared between them highlighted by black atoms and Ag15 atom passing through crystallographic $C_3$ axis highlighted by purple atom; **d** the distribution of three 1D arcuate $[(TC4A)_2(V_4O_9)]^{6-}$ metalloligands in **Ag43**, the purple shell represents silver shell; The top view (**e**) and the side view (**f**) of the distribution of $TC4A^{4-}$ in **Ag43**. Color labels: purple, Ag; cyan, V; yellow, S; gray, C; red, O; green, Cl; blue, N; cyan polyhedron, POVs; blue cup, $TC4A^{4-}$.

well as higher charge density, as confirmed by the bond valence sum (BVS) calculation[29]. The Ag···Ag interactions in **Ag14** and **Ag43** are both around 2.91–3.35 Å, which is larger than the sum of the Ag atom radii (2.89 Å) and shorter than the sum of the van der Waals radii (3.44 Å), indicating the oxidation state of silver is +1 rather than 0[30,31]. Further-more, we also demonstrated that all silver atoms in them are in +1 oxidation state by electrospray ionization mass spectrometry (ESI–MS; see below). The coordination process between $V^{5+}$ cations and $Ag^+$ cations with $TC4A^{4-}$ is speculated to follow the hard-soft acid-base (HSAB) theory, where $V^{5+}$ cations are oxygenophilic and readily coor-dinate to the deprotonated phenolic hydroxyl groups of $TC4A^{4-}$, while $Ag^+$ cations prefer to coordinate with thioether groups. The smaller radius of $V^{5+}$ cations compared with, e.g., Nb, Ta, Mo, and W, allows them to adhere more easily to the bottom of the $TC4A^{4-}$. To the best of

our knowledge, there are only sporadic reports of the high-nuclearity silver NCs protected by metalloligands (Supplementary Table 3). Upon comparison, we find that the mutable forms of POVs in $TC4A^{4-}$–POVs metalloligands provide more variability to their structure, which allowed two structurally different $TC4A^{4-}$–POVs metalloligands to be obtained by adjusting the reaction temperature under otherwise identical conditions.

## Structure transformation between Ag14 and Ag43

ESI–MS is a complementary characterization technique to X-ray crys-tallography for determining the chemical composition and charge state of metal NCs and is widely used to study their solution behavior[32–37]. To investigate the assembly process of **Ag14** and **Ag43**, we monitored the species in the reaction solution at 65–100 °C with

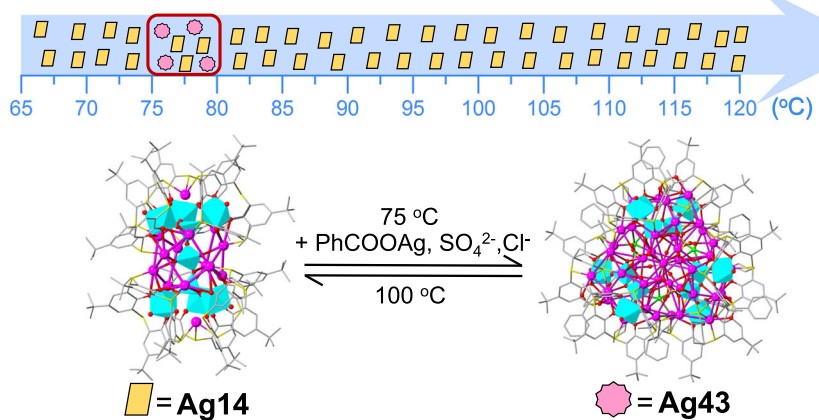

**Fig. 4 | The structure transformation of Ag14 and Ag43.** A schematic of the temperature-dependent assembly process and the structure transformation of **Ag14** and **Ag43**. Color labels: purple, Ag; yellow, S; gray, C; red, O; green, Cl; blue, N; cyan polyhedron, POVs.

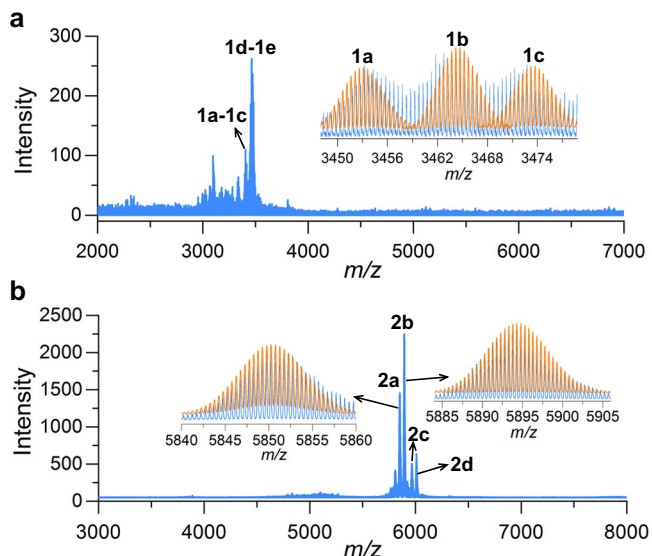

**Fig. 5 | ESI–MS of Ag14 and Ag43.** Positive ion mode ESI–MS of **Ag14** (**a**) and **Ag43** (**b**) dissolved in $CH_2Cl_2$–$CH_3OH$ mixed solvents. Insets: Zoom-in experimental (blue line) and simulated (orange line) isotope patterns of **1a-1c**, **2a**, and **2b** species.

5 °C intervals using ESI–MS while maintaining uniform instrumental test parameters for comparability of data (Supplementary Fig. 12). There were some species correlated to **Ag14** ($Ag_{11}$, $Ag_{14}$, $Ag_{16}$) at 65–100 °C and $Ag_{47}$ species emerged only at 75–80 °C. The latter can be seen as a bigger congener of **Ag43** by appending a PhCOOAg, three $Ag^+$, and solvent molecules. The simulated isotope distributions of **a–f** are shown in Supplementary Fig. 13. These ESI–MS data and synthesis experiments discussed above fully demonstrated that the assembly of **Ag14** and **Ag43** is sensitive to temperature, therefore, there is a promise for the structure transformation of **Ag14** and **Ag43** by thermal induction.

As expected, **Ag14** can be obtained by heating **Ag43** in DMF at 100 °C. However, the transformation from **Ag14** to **Ag43** cannot be achieved by simply changing temperature. By comparing the structures of two NCs, we found the composition of **Ag43** has additional $PhCOO^-$, $SO_4^{2-}$, and $Cl^-$, thus the transformation from **Ag14** to **Ag43** may require the addition of $PhCOO^-$, $SO_4^{2-}$, and $Cl^-$ intentionally except for the change of temperature. We added crystals of **Ag14** to DMF and added an excess of $PhCOO^-$, $SO_4^{2-}$, and $Cl^-$ to facilitate the transformation reaction, and finally obtained **Ag43** by solvothermal

reaction at 75 °C (Fig. 4)[38]. Based on the above results, we believe that the structure transformation between two NCs should suffer from a structure breakage-reorganization route by thermal induction.

In addition to investigating the species in reaction solution at different temperatures by ESI–MS, the solution behavior of **Ag14** and **Ag43** dissolved in $CH_2Cl_2$–$CH_3OH$ mixed solvents was also studied by ESI–MS in positive ion mode. As shown in Fig. 5a, two sets of bivalent charged peaks, **1a–1c** and **1d–1e**, were observed in the $m/z$ range of 2000–7000. **1a–1e** consist of five +2 species, which can be assigned to $[Ag14-2CyS^- + DMF + 2CH_3OH]^{2+}$ (**1a**), $[Ag14-2CyS^- + 2CH_2Cl_2]^{2+}$ (**1b**), $[Ag14-2CyS^- + 2CH_2Cl_2 + H_2O]^{2+}$ (**1c**), $[Ag14 + Ag^+-CyS^- + 2CH_3OH]^{2+}$ (**1d**), and $[Ag14 + Ag^+-CyS^- + 3CH_3OH]^{2+}$ (**1e**), respectively. It can be seen that **1a–1c** species are formed by stripping two $CyS^-$ ligands from **Ag14**, but they still maintain the integrity of 14-nuclei silver framework and 3D $[(V_9O_{16})(TC4A)_6]^{11-}$ metalloligand. All these assigned formulae are listed in Supplementary Table 4. The ESI–MS of **Ag43** has two primary peaks **2a** and **2b**, and both are +2 species (Fig. 5b). Peak **2a** centered at $m/z = 5850.1935$ can be identified to $[Ag43-2PhCOO^- + 2H_2O]^{2+}$ (calcd $m/z = 5850.3182$). Peak **2b** centered at $m/z = 5892.7112$ can be attributed to $[Ag43-2PhCOO^- + CH_2Cl_2 + 2H_2O]^{2+}$ (calcd $m/z = 5892.7942$) (Supplementary Table 5). Both **2a** and **2b** involve the dissociation of the two $PhCOO^-$ ligands in solution but differ in the appended solvent molecules. By analyzing the species in ESI–MS of **Ag14** and **Ag43**, we found that both NCs are Ag(I) clusters and have high stability in $CH_2Cl_2$–$CH_3OH$ mixed solvents.

Of note, the intensity of **1d–1e** species is about 250 at a collision energy of 10 eV, while the intensity of **2b** species is about 2250 without collision energy at the same concentration (Supplementary Table 6). When **Ag14** was tested under the same operating parameters as **Ag43**, no peak emerged (Supplementary Fig. 14). These results indicated that **Ag43** is more easily ionized, which may be related to the labile $PhCOO^-$ on the surface of **Ag43**.

**Photoelectric response properties**

The solid-state UV–Vis diffuse reflectance spectra of **Ag14** and **Ag43** were measured at room temperature in the wavelength range from 200 to 1100 nm (Supplementary Fig. 15). Both of them exhibit a broad absorption spanning ultraviolet and visible regions: one centered at 345 nm and the other at 338 nm. The low-energy broad absorption band is assigned to ligand-to-metal charge transfer, and the high-energy absorption peak is tentatively attributed to the ligand-based absorption. The band gaps of **Ag14** and **Ag43** are 1.78 and 1.70 V, respectively, as determined by the Kubelka−Munk function and Tauc plot[39]. Considering the narrow band gap and wide absorption, the

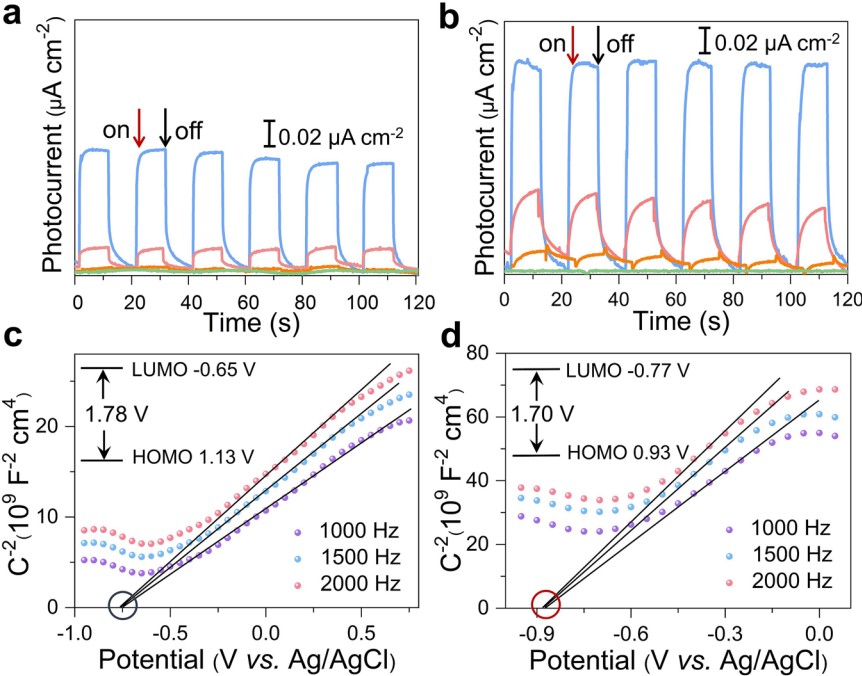

**Fig. 6 | Photocurrent responses of Ag14 and Ag43.** Photocurrent responses of **Ag14** (**a**) and **Ag43** (**b**) modified electrodes under repetitive light irradiation of different wavelengths (blue line, 365 nm; pink line, 420 nm; orange line, 495 nm; green line, 570 nm). Mott–Schottky (M–S) plots of **Ag14** (**c**) and **Ag43** (**d**) modified electrodes at different frequencies.

photoelectrochemical properties of two silver NCs were further tested in a typical three-electrode system[20,39]. With on-off cycling irradiation of different monochrome LED light ($\lambda = 365$, 420, 495, and 570 nm; 50 W; intervals of 10 s), both **Ag14** and **Ag43** exhibit the maximum photocurrent densities under 365 nm irradiation (Fig. 6a, b), and the photocurrent density decreased with increasing wavelength of the irradiation light, which is consistent with their strong absorption at 365 nm. The photocurrent density of **Ag43** (0.1 μA cm$^{-2}$) under 365 nm irradiation was twice that of **Ag14** (0.05 μA cm$^{-2}$), indicating that **Ag43** has better generation and separation efficiency of photoinduced electrons/holes pairs[40].

Mott–Schottky (M–S) measurements were performed using the impedance technique at the frequencies of 1000, 1500, and 2000 Hz to gain a better understanding of the semiconducting nature of **Ag14** and **Ag43** (Fig. 6c, d). The positive slope of the M–S plots proved the *n*-type semiconductor behavior of **Ag14** and **Ag43**[40]. The flat band potential ($E_{FB}$) of *n*-type semiconductors is equal to the Fermi level estimated from the extrapolation of the M–S plots[41]. The conduction band potential (LUMO) of **Ag14** and **Ag43** is −0.65 and −0.77 V vs. normal hydrogen electrode (NHE), respectively. Moreover, the valence band potential (HOMO) of **Ag14** and **Ag43** is calculated to be 1.13 and 0.93 V vs. NHE, respectively, on the basis of the band gap energy obtained from UV–Vis diffuse reflectance spectra.

## Photothermal conversion studies

The energy transition in the photophysical process is mainly illustrated by Jablonski diagram[42], which contains the following several processes: excitation (or absorption), vibrational relaxation (heat), radiative emission (fluorescence), and non-radiative transition (heat) (Supplementary Fig. 16)[43,44]. For **Ag14** and **Ag43**, no fluorescence was observed under 660 nm laser irradiation, indicating that the radiative migration was very weak and photothermal conversion became the main energy release route[45,46]. Therefore, their photothermal conversion performance was investigated both in the crystalline and solution states. As shown in Fig. 7a, the temperature of **Ag14** crystals reached 194 °C in 1.5 s under 660 nm laser irradiation (0.9 W cm$^{-2}$) at a distance of 20 cm.

Surprisingly, the heating rate (115 °C s$^{-1}$) of **Ag14** is much higher than that of other reported silver NC **SD/Ag18a** (8.2 °C s$^{-1}$, 660 nm, 0.9 W cm$^{-2}$) (Supplementary Fig. 17) and most of the reported silver nanomaterials (Supplementary Table 7)[12,47–49]. Compared with **Ag14**, the heating rate of **Ag43** is slower, with the temperature reaching 105 °C in 1.5 s (55.3 °C s$^{-1}$), and the maximum temperature is only 141 °C under the same condition (Fig. 7a). As we have mentioned above, the channels of radiative decay of two NCs are almost suppressed. It is speculated that absorption may be the main factor influencing the photothermal conversion performance, and **Ag14** has stronger absorption at 660 nm than **Ag43** (Supplementary Fig. 15a), therefore, **Ag14** can harvest more energy and produce higher temperature under 660 nm laser irradiation.

The aforementioned results indicate that **Ag14** has the potential to be a remote laser ignition material. The flammable material match was chosen as the research model. The measurement range of the thermal imaging camera is 0–650 °C; when the temperature is higher than 650 °C, it can only display 650 °C. As shown in Fig. 7b, the match coated with 2 mg **Ag14** (hereafter abbreviated as **Ag14/match**) was ignited within 1 s under 660 nm laser irradiation (0.9 W cm$^{-2}$) at a distance of 50 cm, while the match could not be ignited within 3 min under the same condition. Next, the ignition time was investigated by changing the laser power and the irradiation distance, respectively. The **Ag14/match** can be successfully ignited at the laser power range of 0.2 to 0.6 W cm$^{-2}$ at a distance of 20 cm (Fig. 7c). However, the time to ignite the match is much longer than that of **Ag14/match** under the same conditions, and the match cannot be ignited when the laser power is below 0.4 W cm$^{-2}$ (Fig. 7d, e). The time to ignite the **Ag14/match** became longer as the distance changing from 10 cm to 40 cm at laser power of 0.4 W cm$^{-2}$, while match cannot be ignited when the distance exceeds 20 cm (Supplementary Fig. 18, Fig. 7f). On the basis of the above experiments, it can be concluded that the ignition time and the threshold laser power can be significantly reduced by coating the match with **Ag14**, suggesting that **Ag14** has the potential to be used as laser igniter for realizing remote laser ignition and controlled explosion[50–52].

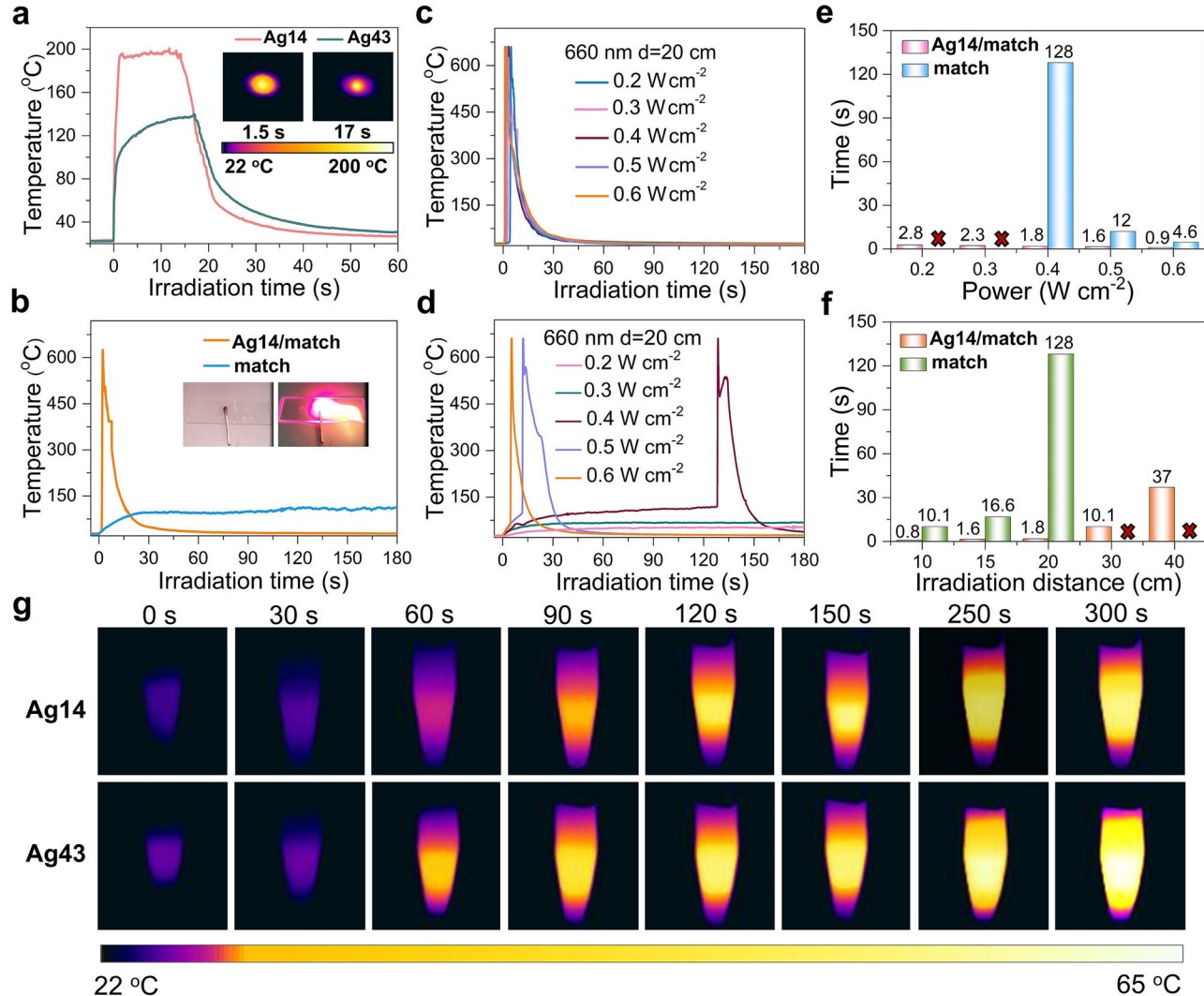

**Fig. 7 | Photothermal conversion performance of Ag14 and Ag43.**
**a** Photothermal conversion of **Ag14** and **Ag43** crystals under 660 nm laser irradiation (0.9 W cm⁻²). Insets: thermal images of **Ag14** and **Ag43** crystals at the highest temperature. **b** The plots of temperature evolution vs. irradiation time for **Ag14**/match and match under 660 nm laser irradiation (0.9 W cm⁻²) at a distance of 50 cm. Insets: photos of **Ag14**/match and ignition. The plots of temperature evolution vs. irradiation time for **Ag14**/match (**c**) and match (**d**) with different laser powers at a distance of 20 cm. The comparison of the ignition time of **Ag14**/match and match at different laser power (**e**) and distance (**f**), the numbers in the figure represent the ignition time, and the red cross mean it cannot be ignited. **g** Thermal images of the CHCl₃ solutions of **Ag14** and **Ag43** at the concentration of 200 μM under 660 nm laser irradiation (0.9 W cm⁻²).

Furthermore, their photothermal conversion performance in solution was investigated. The temperature evolutions of the CHCl₃ solutions of **Ag14** and **Ag43** were recorded by the thermal imaging camera under 660 nm laser irradiation (0.9 W cm⁻²) at concentrations of 50, 100, and 200 μM. The thermal imaging photographs show the photothermal effect of the CHCl₃ solutions of **Ag14** and **Ag43** at the concentration of 200 μM in a centrifuge tube (Fig. 7g). The maximum photothermal temperature of **Ag43** could reach 60.0 °C, which was 3.8 °C higher than that of **Ag14** (56.2 °C) (Supplementary Fig. 19). This may be due to that the UV–Vis absorption intensity of **Ag43** at 660 nm is higher than that of **Ag14** at the same concentration (Supplementary Fig. 20)[53,54]. The photothermal conversion efficiency (η) values of the CHCl₃ solutions of **Ag14** and **Ag43** were 45.69% and 33.87%, respectively, at a concentration of 200 μM under 660 nm laser irradiation (Supplementary Figs. 21 and 22)[55]. The CHCl₃ solution of **Ag14** has a high η compared to the reported four Ag₁₆ NCs (41.1%, 35.8%, 40.7%, and 33.6%) as well as dopamine-melanin colloidal nanospheres (40%)[47,56]. Moreover, the maximum temperatures increased with increasing concentrations of **Ag14** and **Ag43** and remained almost constant in five cycles of the heating and

cooling process (Supplementary Fig. 23). These results indicated that the solutions of the two silver NCs have good photothermal conversion performance, making them potentially applicable in bioimaging and photothermal therapy. The compared UV–Vis spectra of fresh samples and samples stored in ambient conditions for 6 months and the photothermal heating and cooling process confirmed that **Ag14** and **Ag43** exhibit high stability to light and air both in the solid and solution states (Supplementary Figs. 15a and 24).

## Discussion

In summary, we showcase an efficient strategy to synthesize two silver NCs consolidated by 3D scaffold-like [(TC4A)₆(V₉O₁₆)]¹¹⁻ and 1D arcuate [(TC4A)₂(V₄O₉)]⁶⁻ metalloligands. Although there has been extensive research on organic–inorganic hybrid POVs, this work represents the study targeting two organic ligand-modified POVs as metalloligands for the stabilization of silver NCs utilizing multiple-site and multiple-dentate (MSMD) coordination patterns. On the other hand, TC4A⁴⁻-POVs hybrids with dual functionalities as internal anion template and external ligand have been proven to be reliable ligands for constructing multinuclear silver NCs. Noteworthy, the exceptional

photothermal conversion performance of **Ag14** makes it a promising material for remote laser ignition. The design and synthesis of metalloligands present exciting challenges and offer a fertile platform for exploration, opening up opportunities for creativity in combining the esthetics of synthetic chemistry with self-assembly, which has the potential to drive further developments in the synthesis of silver NCs. Our research expands the scope of assembling macrocyclic thiacalix[4] arene ligands with POVs in a sensible manner and provides an avenue for further design and understanding metalloligands.

## Methods

### Synthesis of (CySAg)$_n$

(CySAg)$_n$ was synthesized by the following reported procedure[57]. Firstly, AgNO$_3$ (30 mmol, 5 g) was dissolved in 75 mL MeCN, and CySH (30 mmol, 3.66 mL) and Et$_3$N (36 mmol, 5 mL) were added into 100 mL EtOH. Then, the above two solutions were mixed and stirred for 5 h in the dark, and the light yellow powdery (CySAg)$_n$ was obtained with a yield of 90% (based on AgNO$_3$) by filtration of the above light yellow suspension. Selected IR peaks (cm$^{-1}$): 2920 (s), 1435 (m), 1269 (m), 998 (m), 719 (m) (Supplementary Fig. 25).

### Synthesis of H$_4$TC4A

H$_4$TC4A was synthesized by the following reported procedure[58]. Firstly, the mixture of *p*-tert-butylphenol (0.43 mol, 64.5 g), S$_8$ (0.86 mol, 27.5 g), and NaOH (0.215 mol, 8.86 g) was added to a 500 mL flask containing tetraethylene glycol dimethyl ether (19 mL) under nitrogen atmosphere. Then, the temperature was raised to 230 °C within 240 min and held for 180 min. The resulting hydrogen sulfide is removed by a slow flow of nitrogen during the reaction. After cooling to room temperature, ether (140 mL) and toluene (35 mL) were added to the flask to dilute the dark red product, and then the above solution was acidified with 4 M H$_2$SO$_4$ (140 mL) for 1 h. The precipitate was collected by filtration and recrystallized from CHCl$_3$ with a yield of 20% (based on *p*-tert-butylphenol). $^1$H NMR (400 MHz, CDCl$_3$): δ = 9.60 (s, 4H, OH), 7.63 (s, 8H, Ar-H), 1.22 (s, 36H, C(CH$_3$)$_3$). Selected IR peaks (cm$^{-1}$): 3334 (m), 2962 (m), 1450 (s), 1397 (m), 1241 (s), 880 (m), 740 (s) (Supplementary Fig. 26).

### Synthesis of Ag2

The synthesis of needle-black crystals of **Ag2** involves the following steps. Firstly, H$_4$TC4A (0.015 mmol, 10.8 mg), VOSO$_4$·$x$H$_2$O (0.05 mmol, 8 mg), and PhCOOAg (0.02 mmol, 4.6 mg) were dispersed in 1.5 mL DMF and the suspension was stirring (800 rpm) for 6 h at room temperature (20 °C). Then the mixture was sealed in a 25 mL Teflon-lined stainless autoclave and heated at 65 °C for 33 h. Black needle-like crystals of **Ag2** were collected with a yield of 15% after the solvothermal reaction (based on PhCOOAg). Elemental analyses calc. (found) for **Ag2** (C$_{89}$H$_{145}$Ag$_2$N$_3$O$_{31}$S$_8$V$_2$): C, 45.89 (47.68); H, 6.23 (5.586); N, 1.80 (1.66)%. Selected IR peaks (cm$^{-1}$): 2951 (m), 2348 (m), 1647 (s), 1424 (s), 1251 (m), 1075 (m), 983 (m), 833 (m), 757 (m), 538 (m).

### Synthesis of Ag14

Typically, H$_4$TC4A (0.015 mmol, 10.8 mg), VOSO$_4$·$x$H$_2$O (0.05 mmol, 8 mg), and PhCOOAg (0.1 mmol, 22.9 mg) were dissolved in 1.5 mL DMF. After stirring (800 rpm) at room temperature (20 °C) for 3 h, (CySAg)$_n$ (0.05 mmol, 11.4 mg) and NaCl (0.017 mmol, 1 mg) were added into above solution which is further treated for 3 h under the same stirring (800 rpm) condition. Then the mixture was sealed in a 25 mL Teflon-lined stainless autoclave and heated at 65 °C for 33 h. After cooling to room temperature, black clump-like crystals of **Ag14** were formed with a yield of 15% (based on H$_4$TC4A). The above synthetic reaction can readily achieve a tenfold scale-up and produce 20.6 mg **Ag14** in one batch. Elemental analyses calc. (found) for **Ag14** (C$_{273}$H$_{372}$Ag$_{14}$N$_5$O$_{65}$S$_{27}$V$_9$): C, 43.11 (40.72); H, 4.89 (4.311); N, 0.92

(0.94)%. Selected IR peaks (cm$^{-1}$): 2953 (m), 2366 (m), 1678 (m), 1419 (s), 1260 (s), 1075 (m), 886(m), 821 (s), 758 (s), 686(s), 531(s).

### Synthesis of Ag43

The synthesis of **Ag43** was similar to those described for **Ag14**. The only difference is the solvothermal reaction temperature, with the mixture of **Ag14** and **Ag43** being obtained between 75 and 80 °C at the solvothermal reaction. **Ag43** is a black block-like crystal with a yield of 7%, and **Ag14** with a yield of 10% (based on H$_4$TC4A). Elemental analyses calc. (found) for **Ag43** (C$_{342}$H$_{485}$Ag$_{43}$Cl$_3$N$_9$O$_{104}$S$_{38}$V$_{12}$): C, 31.66 (30.44); H, 3.70 (3.758); N, 0.97 (0.84)%. Selected IR peaks (cm$^{-1}$): 2950 (m), 2362 (m), 1680 (s), 1416 (s), 1262 (m), 1076(m), 803(s), 708(s), 537 (m).

## Data availability

The data that support the findings of this study are available within the article and its Supplementary Information files. Other relevant data are available from the corresponding author upon request. The X-ray crystallographic coordinates for structures reported in this article have been deposited at the Cambridge Crystallographic Data Center under deposition numbers CCDC: 2251154, 2251155, and 2251156 for **Ag2**, **Ag14**, and **Ag43**. These data can be obtained free of charge from the Cambridge Crystallographic Data Center via www.ccdc.cam.ac.uk/data_request/cif.

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

## Acknowledgements

This work was financially supported by the National Natural Science Foundation of China (Grant Nos. 22201159 to Z.W., 22171164, 22325105, 52261135637 to D.S.), the Natural Science Foundation of Shandong Province (No. ZR2022QB008 to Z.W.), the National Postdoctoral Innovative Talents Support Program (No. BX2021171 to Z.W.), China Postdoctoral Science Foundation (No. 2021M700081 to Z.W.) and the Instrument Improvement Funds of Shandong University Public Technology Platform (ts20220102).

## Author contributions

The original idea was conceived by D.S., experiments and data analyses were performed by Z.W., Y.-J.Z., B.-L.H., Y.-Z.L., C.-H.T., and D.S., ESI-MS data were collected by B.-L.H. and Y.-J.Z., structure characterization was performed by Z.W., Y.-J.Z., and D.S., the paper was drafted by D.S., Z.W., and Y.-J.Z. All authors have given approval to the paper.

## Competing interests

The authors declare no competing interests.
