## [Peer Review File · Nature Communications]

A Route to Metalloligands Consolidated Silver Nanoclusters by Grafting Thiocalix[4]arene onto PolyoxovanadatesReviewers' Comments:

Reviewer #1:

Remarks to the Author:

In the paper, Sun group once again presents important advances in silver nanoclusters, reporting the synthesis and professional, top-level characterization of two novel silver nanoclusters protected by TC4A4--POVs metalloligands. In my opinion, such a synthetic strategy could only be proposed after intensive research in this field, which revealed 3D scaffold-like and 1D arcuate TC4A4--POVs metalloligands with a completely new structure and a dual functional role. The photothermal conversion of the two clusters has been fully studied, and it is found that the solutions of Ag14 and Ag43 have high photothermal conversion efficiency. Besides, the high plateau temperature and the fast heating rate of the Ag14 in solid state have been fully utilized, and match as a simple model has been chosen to visualize the great potential of Ag14 as a photothermal material. Overall, this study not only provides important insights into designing metalloligands, but also promotes the application of silver nanoclusters in the field of photothermal conversion materials. This work will be of interest to the readers of Nat. Commun and deserves publication. I would like to suggest accepting this paper after addressing minor revisions listed below.

1. As a photothermal material, whether Ag14 can be synthesized in macroscopic quantities?
2. In the synthesis method section, the yield of Ag2 Ag14 and Ag43 needs to include the basis for yield calculations.
3. It is well known that silver nanoclusters are sensitive to light and less stable compared to gold nanoclusters. Are these two cases of silver nanoclusters stable in solid and solution states in air?
4. The author used VOSO4 in the synthesis of Ag14, but there is no SO4²⁻ in the structure of Ag14. Is it possible to synthesize Ag14 using other vanadium sources?
5. Please double check the format of refs. 30 and 48.
6. The caption of scheme 1 is inaccurate and should be a schematic diagram of the assembly of the TC4A4--POVs metalloligands.
7. Author should indicate the meanings represented by the different colored atoms in Fig 2c.
8. These literatures should be cited in photothermal conversion section: J. Phys. Chem. C 2007, 111, 9, 3636-3641; Int. Rev. Phys. Chem. 2000, 19, 3, 409-453.
9. There are some inappropriate wordings in the manuscript. Please polish or proofread.

Reviewer #2:

Remarks to the Author:

The authors have reported two new silver NCs, {Ag14[(TC4A)6(V9O16)](CyS)3} (Ag14) and {Ag43S[(TC4A)2(V4O9)]3(CyS)9(PhCOO)3Cl3(SO4)4(DMF)3·6DMF} (Ag43), by controlling the reaction temperature. Ag14 shows superior photothermal conversion performance compared with Ag43 in the solid state. This work may potentially interest readers of Nature Communications working in the area of polyoxometalate and NC chemistry. The authors should answer the following comments to meet the standards for publication in Nature Communications.

- (1)The oxidation state of vanadium is mentioned, but the oxidation state of silver seems to be missing.
- (2)Elemental analysis of the compounds should be given.
- (3)I understand that this is the first report of the combination of thiacalix[4]arene and POV into a metalloligand entity. However, I am unable to comprehend the "dual benefit" of this combination. The phrase "leading to the emergence of unexpected assembly phenomena that cannot be realized by using organic ligands alone (lines 62-64)" is unclear, considering that "template effects," "stability enhancement," and "prevention of further aggregation" by ligands have been well known for a long time.
- (4)The discussion part is too brief. It would be beneficial to include a concise perspective on the future challenges of Ag clusters with metalloligands.
- (5)The structures and functions of Ag14 and Ag43 should be compared with similar systems to provide

a better understanding of the novelty of these clusters.

(6) The authors state that the structural conversion from Ag₄₃ to Ag₁₄ can be achieved by adjusting the temperature, while the reverse reaction needs stimuli from both temperature and reactant additives. Why is the transformation process irreversible?

Reviewer #3:

Remarks to the Author:

This manuscript reports the stabilization of two novel Ag nanoclusters by metalloligands resulting from the combination of a thiacalix[4]arene and polyoxovanadates. Such a combination is unprecedented and the resulting structures are actually particularly original. Furthermore, owing to the probable variability in the way the thiacalixarene and the polyoxovanadate proligands can assemble, this work opens the door for a likely rich and innovative chemistry of nanoclusters protected by metalloligands. This is all the more interesting as the reported compounds show exceptional performances in photothermal conversion. This outstanding work deserves publication in Nature Comm., providing minor modifications as detailed below.

1) The Ag coordination spheres would deserve some description.

2) Is there any rational explanation for the outstanding (but somewhat unexpected) photothermal properties of compound Ag₁₄?

3) In these compounds, Ag is in the +I oxidation state. By using a reducing agent stronger than DMF, would it be possible to generate partly reduced (Ag)_n nanoclusters, or perhaps vanadium(+IV) centers, with the possibility for electron transfer between metals?

Reviewer #1 (Remarks to the Author):

In the paper, Sun group once again presents important advances in silver nanoclusters, reporting the synthesis and professional, top-level characterization of two novel silver nanoclusters protected by TC4A⁴⁻-POVs metalloligands. In my opinion, such a synthetic strategy could only be proposed after intensive research in this field, which revealed 3D scaffold-like and 1D arcuate TC4A⁴⁻-POVs metalloligands with a completely new structure and a dual functional role. The photothermal conversion of the two clusters has been fully studied, and it is found that the solutions of Ag14 and Ag43 have high photothermal conversion efficiency. Besides, the high plateau temperature and the fast heating rate of the Ag14 in solid state have been fully utilized, and match as a simple model has been chosen to visualize the great potential of Ag14 as a photothermal material. Overall, this study not only provides important insights into designing metalloligands, but also promotes the application of silver nanoclusters in the field of photothermal conversion materials. This work will be of interest to the readers of *Nat. Commun* and deserves publication. I would like to suggest accepting this paper after addressing minor revisions listed below.

Response: We are very pleased and excited by the positive comments on the novelty and significance of our study. We are also very grateful for the reviewer's comments and questions in helping us improve the manuscript.

1. As a photothermal material, whether Ag14 can be synthesized in macroscopic quantities?

Response: Thanks for your important comment. Scale-up synthesis is crucial for the exploration of the properties and practical applications. However, the assembly process of metal nanoclusters is sensitive to synthesis conditions, making it difficult to achieve high-yield synthesis. Fortunately, scale-up synthesis of Ag14 can be achieved by expanding the amount of reactants and solvent by a factor of 10. The following discussion was added to the Methods section of the main text: "The above synthetic reaction can readily achieve a tenfold scale-up and produced 20.6 mg Ag14 in one batch."

2. In the synthesis method section, the yield of Ag2 Ag14 and Ag43 needs to include

the basis for yield calculations.

Response: Thanks for your important reminder on the calculation of the yield of three compounds. We calculated yield based on the insufficient reactant: Ag2 (yield: 15 %, based on PhCOOAg), Ag14 (yield: 15 %, based on H₄TC4A) and Ag43 (yield: 7 %, based on H₄TC4A). We supplemented the yield calculation basis in the Methods section of the main text.

3. It is well known that silver nanoclusters are sensitive to light and less stable compared to gold nanoclusters. Are these two cases of silver nanoclusters stable in solid and solution states in air?

Response: Thanks for your constructive suggestion. Some silver nanoclusters have poor stability when exposed to air or light. However, Ag14 and Ag43 can maintain color and morphology unchanged in the environment for six months in the solid state. The compared UV-Vis spectra of fresh samples and samples stored in ambient conditions for six months supported the above conclusion and were added in Supplementary Fig. 15a (same as Figure R1). Besides, the UV-Vis spectra of the CHCl₃ solutions of Ag14 and Ag43 remain basically unchanged after five photothermal heating and natural cooling cycles under 660 nm irradiation (Supplementary Fig. 24). The above-mentioned results demonstrate that Ag14 and Ag43 exhibit high stability to light and air both in the solid and solution states.

Figure R1: The UV-Vis spectra of fresh samples and samples stored in ambient conditions for six months (a), insets: photographs of Ag14 and Ag43. Diffuse reflectance UV-Vis spectra of Kubelka-Munk function vs energy and Tauc plots of Ag14 (b) and Ag43 (c).

Furthermore, we added the discussion for the stability of Ag14 and Ag43 in the main text: "The compared UV-Vis spectra of fresh samples and samples stored in ambient conditions for six months and photothermal heating and cooling process confirmed that Ag14 and Ag43 exhibit high stability to light and air both in the solid and solution states (Supplementary Fig. 15a and Supplementary Fig. 24).".

4. The author used VO_2 in the synthesis of Ag14, but there is no SO_4^{2-} in the structure of Ag14. Is it possible to synthesize Ag14 using other vanadium sources?

Response: Thanks for your constructive suggestion. According to the reviewer's suggestion, we used the alternative vanadium reagents, such as NaVO_3 or Na_3VO_4 that are available in our laboratory for the synthesis of Ag14 under otherwise identical conditions. Fortunately, the crystals of Ag14 were also obtained after the solvothermal reaction, indicating a non-specificity of the vanadium source in the synthesis of Ag14. In addition, we added the following to the synthesis discussion section: "In addition, the vanadium source in the synthesis of Ag14 is non-specific, as evidenced by the successful crystallization of Ag14 after solvothermal reaction using other vanadium reagents, such as NaVO_3 or Na_3VO_4 under otherwise conditions.".

5. Please double check the format of refs. 30 and 48.

Response: Thanks for your careful checking. We have carefully checked the format of the refs. 30 and 48 and corrected these errors.

6. The caption of scheme 1 is inaccurate and should be a schematic diagram of the assembly of the TC4A^{4-} -POVs metalloligands.

Response: Thanks for your careful inspection. We have carefully considered your suggestion and modified the caption of scheme 1 as "Schematic diagram of the assembly of TC4A^{4-} -POVs metalloligands.". Furthermore, we have revised Scheme 1 to Fig. 1 according to the formatting instructions.

7. Author should indicate the meanings represented by the different colored atoms in Fig 2c.

Response: Thanks for your constructive suggestion. We are sorry for the confusion caused by the lack of color instructions. All the colored atoms in Fig 2c are silver atoms, and the purpose of using different colors is to distinguish different parts of

the silver shell. We explain in the legend: "The Ag₄₃ shell in (b), each Ag₁₅ caps highlighted individually by different colors (pink, orange and green), silver atoms shared between them highlighted by black atoms and Ag₁₅ atom passing through crystallographic C₃ axis highlighted by purple atom;". In addition, we have revised Fig. 2 to Fig. 3 according to the formatting instructions.

8. These literatures should be cited in photothermal conversion section: J. Phys. Chem. C 2007, 111, 9, 3636-3641; Int. Rev. Phys. Chem. 2000, 19, 3, 409-453.

Response: Thanks for your constructive suggestion. These two literatures are very helpful for our in-depth understanding of photothermal conversion performance and calculating photothermal conversion efficiency. We cited these two important literatures in the References section of the main text:"

References:

45. Roper, D. K., Ahn, W. & Hoepfner, M. Microscale heat transfer transduced by surface plasmon resonant gold nanoparticles. J. Phys. Chem. C **III**, 3636-3641 (2007).

46. Link, S. & El-Sayed, M. A. Shape and size dependence of radiative, non-radiative and photothermal properties of gold nanocrystals. Int. Rev. Phys. Chem. **19**, 409-453 (2000)."

9. There are some inappropriate wordings in the manuscript. Please polish or proofread.

Response: Thank you for finding these. We have carefully checked the overall manuscript and removed the typos.

Reviewer #2 (Remarks to the Author):

The authors have reported two new silver NCs, {Ag₁₄[(TC4A)₆(V₉O₁₆)](CyS)₃} (Ag14) and {Ag₄₃S[(TC4A)₂(V₄O₉)]₃(CyS)₉(PhCOO)₃Cl₃(SO₄)₄(DMF)₃·6DMF} (Ag43), by controlling the reaction temperature. Ag14 shows superior photothermal conversion performance compared with Ag43 in the solid state. This work may potentially interest readers of Nature Communications working in the area of polyoxometalate and NC chemistry. The authors should answer the following comments to meet the standards for publication in Nature Communications.

Response: We are very pleased and excited by the positive comments on the novelty and significance of our study. We would also like to thank the reviewer for his/her inspiring and constructive comments/suggestions, which have been taken into careful consideration in this revision. We believe that the quality of the manuscript has been improved thanks to your constructive suggestions.

(1) The oxidation state of vanadium is mentioned, but the oxidation state of silver seems to be missing.

Response: Thanks for your constructive suggestion. Single crystal X-ray diffraction (SCXRD) is a powerful and directed method that allows for detailed structural information at the atomic level, *ie.*, every atom (except H) can be identified and located with high accuracy. From the SCXRD results, the metal atom arrangements, metal-ligand interface structure, and the distance between two metal atoms or between a metal atom and a ligand can be understood. For Ag14 and Ag43, the Ag...Ag interactions in them are both around 2.91-3.35 Å, which is larger than the sum of the metal atom radii (2.89 Å) and shorter than the sum of the van der Waals radii (3.44 Å), indicating the oxidation state of silver is +1 rather than 0 (*Angew. Chem. Int. Ed.* 2015, 54, 746-784.). As a complementary of SCXRD, electrospray mass spectrometry (ESI-MS) is another powerful method for determining the molecular formula, charge state, solution behavior, and assembly mechanism of metal NCs (*J. Am. Chem. Soc.* 2016, 138, 1328-1334; *Angew. Chem. Int. Ed.* 2020, 59, 5312-5315; *Angew. Chem. Int. Ed.* 2020, 59, 17234-17238). Then, we can infer the oxidation state of silver from ESI-MS result. For

Ag14, taking 1b for example, with the formula consistent with $\{\text{Ag}_{14}[(\text{TC4A})_6(\text{V}_9\text{O}_{16})](\text{CyS})(\text{CH}_2\text{Cl}_2)_2\}^{2+}$, the oxidation state of Ag is calculated to be +1 ($+1 = \{+2 \text{ (charge state of 1b)} - [-1 (\text{CyS}^-)] - \{-11 [(\text{TC4A})_6(\text{V}_9\text{O}_{16})]^{11-}\} / 14$), which inversely confirms the correctness of the SCXRD result. In addition, the +1 oxidation state of Ag is also suitable for other labeled species in ESI-MS of Ag14. Similarly, it can be inferred that all silver are also in +1 oxidation state in Ag43. In light of the above observations, the oxidation state of Ag in both Ag14 and Ag43 is +1.

In addition, we add the description of the silver oxidation state to X-ray crystal structures and ESI-MS sections as: "*The Ag...Ag interactions in Ag14 and Ag43 are both around 2.91-3.35 Å, which is larger than the sum of the Ag atom radii (2.89 Å) and shorter than the sum of the van der Waals radii (3.44 Å), indicating the oxidation state of silver is +1 rather than 0.^{30,31} Furthermore, we also demonstrated that all silver atoms in them are in +1 oxidation state by electrospray ionization mass spectrometry (ESI-MS; vide infra).*" and "*By analyzing the species in ESI-MS of Ag14 and Ag43, we found that both NCs are Ag(I) cluster and have high stability in CH₂Cl₂-CH₃OH mixed solvents.*".

Furthermore, the related references were also listed in the "References" section of the main text:"

References:

30. Schmidbaur, H. & Schier, A. Argentophilic Interactions. Angew. Chem. Int. Ed. 54, 746-784 (2015).

31. Pyykkö, P. Strong closed-shell interactions in inorganic chemistry. Chem. Rev. 97, 597-636 (1997)."

(2) Elemental analysis of the compounds should be given.

Response: Thanks for your constructive comment. According to the reviewer's suggestion, we carried out elemental analysis tests for three compounds (Ag2, Ag14 and Ag43) and found that the experimental values are consistent with the calculated values obtained from the SCXRD.

Furthermore, we added elemental analysis to the synthesis section: "*Elemental*

analyses calc. (found) for Ag2 (C₈₉H₁₄₅Ag₂N₃O₃₁S₈V₂): C, 45.89 (47.68); H, 6.23 (5.586); N, 1.80 (1.66)%. Elemental analyses calc. (found) for Ag14 (C₂₇₃H₃₇₂Ag₁₄N₅O₆₅S₂₇V₉): C, 43.11 (40.72); H, 4.89 (4.311); N, 0.92 (0.94)%. Elemental analyses calc. (found) for Ag43 (C₃₄₂H₄₈₅Ag₄₃Cl₃N₉O₁₀₄S₃₈V₁₂): C, 31.66 (30.44); H, 3.70 (3.758); N, 0.97 (0.84)%.

(3) I understand that this is the first report of the combination of thiacalix[4]arene and POV into a metalloligand entity. However, I am unable to comprehend the “dual benefit” of this combination. The phrase “leading to the emergence of unexpected assembly phenomena that cannot be realized by using organic ligands alone (lines 62-64)” is unclear, considering that “template effects,” “stability enhancement,” and “prevention of further aggregation” by ligands have been well known for a long time.

Response: Thanks for your constructive suggestion. The dual benefits of TC4A⁴⁻-POVs metalloligands are that the surface thiacalix[4]arenes act as multi-dentate ligands to protect the cluster, and the internal POVs serve as an anionic template. The phrase "leading to the emergence of unexpected assembly phenomena that cannot be realized by using organic ligands alone (lines 62-64 in the main text)" is intended to convey that TC4A⁴⁻-POVs metalloligands may lead to richer coordination patterns and assembly phenomena than organic ligands. The phrase may cause ambiguity and we replaced it with: "Herein, we envisioned that integrating thiacalix[4]arenes and POVs into an entity can provide the dual function of multi-dentate chelating of thiacalix[4]arenes and anionic templating of POVs, leading to the emergence of richer coordination patterns and assembly phenomena than using organic ligands.". We hope that the revised sentence will express this dual function more clearly.

(4) The discussion part is too brief. It would be beneficial to include a concise perspective on the future challenges of Ag clusters with metalloligands.

Response: Thanks for your constructive suggestion. We added the following discussions in the discussion part: "The design and synthesis of metalloligands present exciting challenges and offer a fertile platform for exploration, opening up opportunities for creativity in combining the aesthetics of synthetic chemistry with self-

assembly, which has the potential to drive further developments in the synthesis of silver NCs."

(5) The structures and functions of Ag14 and Ag43 should be compared with similar systems to provide a better understanding of the novelty of these clusters.

Response: Thanks for your constructive suggestion. The structural and functional novelties of Ag14 and Ag43 are the TC4A⁴⁻-POVs metalloligands and photothermal conversion performance. On the basis of your suggestion, we compare the TC4A⁴⁻-POVs metalloligands and photothermal conversion performance of Ag14 and Ag43 with reported similar systems. Recently, metalloligands can be effective candidates for constructing functional metal-organic frameworks/cages, but are rarely used for constructing silver NCs. To the best of our knowledge, there are only sporadic reports of the high-nuclearity silver NCs protected by metalloligands (Table R1, same as Supplementary Table 3). Among these reports, we find that the mutable forms of POVs in TC4A⁴⁻-POVs metalloligands provide more variability to their structure. The unique characteristic of TC4A⁴⁻-POVs metalloligands lie in their dual-functional role, with the TC4A⁴⁻ on the surface as protective ligands and the inner POVs as anionic templates. The photothermal conversion studies of metalloligand-protected silver NCs are still rare and need to be further investigated. Therefore, we compared the photothermal properties of Ag14 and Ag43 with other silver nanomaterials and found that Ag14 has a fast heating rate (Table R2, same as Supplementary Table 7), which can effectively reduce the time delay as a remote laser ignition material. We summarize the reported high-nuclearity silver NCs protected by metalloligands and the photothermal conversion performance of silver nanomaterials in solid-state in Supplementary Table 3 and Supplementary Table 7, respectively, and added the following discussions into the relevant position of main text: "*To the best of our knowledge, there are only sporadic reports of the high-nuclearity silver NCs protected by metalloligands (Supplementary Table 3). Upon comparison, we find that the mutable forms of POVs in TC4A⁴⁻-POVs metalloligands provide more variability to their structure, which allowed two structurally different*

TC4A⁴⁻-POVs metalloligands to be obtained by adjusting the reaction temperature under otherwise identical conditions." and "Surprisingly, the heating rate (115 °C s⁻¹) of Ag14 is much higher than that of other reported silver NC SD/Ag18a (8.2 °C s⁻¹, 660 nm, 0.9 W cm⁻²) (Supplementary Fig. 17) and most of the reported silver nanomaterials (Supplementary Table 7)."

Table R1: The summary of metalloligand-protected silver nanoclusters (NCs).^{9,14-}

17

Metalloligand formula	Metalloligand structure	Silver NC formula	Silver NC structure	Ref.
TiL ₃ (L= salicylate or 5-fluorosalicylate)		Ti ₄ Ag ₈ (SA) ₁₂ (H ₂ SA = salicylic acid)		14
		Ti ₄ Ag ₁₂ (S'Pr) ₆ (SA) ₁₀ (HSA) ₂		
		Ti ₄ Ag ₂₂ (S'Pr) ₁₂ (SA) ₁₂ SO ₄		
		Ti ₄ Ag ₄₂ (S) ₄ (S'Pr) ₁₈ (SA) ₁₂ (SO ₄) ₄		
	Ti ₄ Ag ₃₆ (S'Pr) ₂₄ (SA-F) ₁₂ (SO ₄) ₂ (H ₂ SA-F = 5-fluorosalicylic acid)			

$[\text{Mo}_2\text{O}_5(\text{PTC4A})_2]^{6-}$ (H ₄ PTC4A = p -phenylthiacalix[4]arene)		$[\text{Ag}_{18}(\text{Mo}_2\text{O}_5\text{PTC4A})_2(\text{EtS})_6(\text{Tos})_2] \cdot 2\text{Ag}(\text{C}_6\text{H}_5\text{CN})_3$		9
		$\text{Ag}_{18}\text{S}\{\text{Mo}_2\text{O}_5(\text{PTC4A})_2[\text{MoO}_2(^i\text{PrO})][\text{MoO}(^i\text{PrO})_2]\}_2(\text{CyS})_6(\text{Tos})_2(^i\text{PrO})_2$		
$\text{MoO}_3\text{-TC4A}$ (H ₄ TC4A = p -tert-butylthiacalix[4]arene)		$\text{NH}_4[\text{Cl}@\text{Ag}_{42}(\text{MoO}_3\text{-TC4A})_6(\text{EtS})_{18}]$		15
$\text{Rh}(\text{aet})_3$ (Haet = 2-aminoethanethiolate)		$[\text{Ag}_{46}\text{S}_{13}\{\text{Rh}(\text{aet})_3\}_{14}]^{20+}$		16
$\text{Rh}(\text{apt})_3$ (Hapt = 3-aminopropanethiol)		$[\text{Ag}_{11}\text{S}\{\text{Rh}(\text{apt})_3\}_6]^{9+}$		17
		$[\text{Ag}_{13}\text{S}\{\text{Rh}(\text{apt})_3\}_6]^{11+}$		
$\text{TC4A}^+\text{-POVs}$ (H ₄ TC4A = p -tert-butylthiacalix[4]arene)		$\text{Ag}_{14}[(\text{TC4A})_6(\text{V}_9\text{O}_{16})_3(\text{CyS})_3]$		This work

ne, POVs = polyoxovanadates)		Ag ₄₃ S[(TC4A) ₂ (V ₄ O ₉) ₃ (CyS) ₉ (PhCOO) ₃ Cl) ₃ (SO ₄) ₄ (DMF) ₃ ·6DMF		
---	--	--	--

Table R2: Reported solid-state photothermal silver nanomaterials.^{9,11,18,19}

Compound	Laser power (W cm ⁻²)	Laser wavelength (nm)	Heating rate (°C s ⁻¹)	Maximum temperature (°C)	Irradiation distance (cm)	Ref.
Ag16(I)	0.1	420-780	8.16	81.6	/	11
Ag16(II)			7.23	72.3		
Ag16(III)			7.04	70.4		
Ag16(IV)			7.03	70.3		
Ag nanoparticle	10 ⁵	530	/	80.7	/	18
Ag nanotriangle		740		258.5		
Ag nanorod		800		294.3		
Ag NPs@MOF	0.7	808	13.2	239.8	8	19
SD/Ag18a	0.9	660	8.2	187	20	9
Ag14	0.9	660	115	194	20	This work
Ag43			70	105		

Furthermore, the related references were also listed in the "Supplementary References" section of the Supplementary Information:"

Supplementary References:

14. Gao, M.-Y. et al. *Tetrahedral geometry induction of stable Ag-Ti nanoclusters by flexible trifurcate TiL₃ metalloligand. J. Am. Chem. Soc.* **142**, 12784-12790 (2020).
15. Wang, Z. et al. *Stepwise assembly of Ag₄₂ nanocalices based on a Mo^{VI}-anchored thiacalix[4]arene metalloligand. ACS Nano* **16**, 4500-4507 (2022).
16. Ueda, M., Goo, Z. L., Minami, K., Yoshinari, N. & Konno, T. *Structurally precise*

silver sulfide nanoclusters protected by rhodium(III) octahedra with aminothiulates. Angew. Chem. Int. Ed. 58, 14673-14678 (2019).

17. Yoshinari, N., Goo, Z. L., Nomura, K. & Konno, T. Silver(I) sulfide clusters protected by rhodium(III) metalloligands with 3-aminopropanethiolate. Inorg. Chem. 62, 9291-9294 (2023).

18. Borah, R. & Verbruggen, S. W. Silver-gold bimetallic alloy versus core-shell nanoparticles: implications for plasmonic enhancement and photothermal applications. J. Phys. Chem. C 124, 12081-12094 (2020).

19. Su, J. et al. Enhancing the photothermal conversion of tetrathiafulvalene-based MOFs by redox doping and plasmon resonance. Chem. Sci. 13, 1657-1664 (2022)."

(6) The authors state that the structural conversion from Ag43 to Ag14 can be achieved by adjusting the temperature, while the reverse reaction needs stimuli from both temperature and reactant additives. Why is the transformation process irreversible?

Response: Thanks for your constructive suggestion. By comparing the molecular formulae and structures of Ag14 ($\{\text{Ag}_{14}[(\text{TC4A})_6(\text{V}_9\text{O}_{16})](\text{CyS})_3\}$) and Ag43 ($\{\text{Ag}_{43}\text{S}[(\text{TC4A})_2(\text{V}_4\text{O}_9)]_3(\text{CyS})_9(\text{PhCOO})_3\text{Cl}_3(\text{SO}_4)_4(\text{DMF})_3 \cdot 6\text{DMF}\}$), we find that the component of Ag43 are more complex than that of Ag14, with PhCOO^- , SO_4^{2-} and Cl^- absent in Ag14. According to the law of conservation of mass, the transformation from Ag14 to Ag43 requires the addition of the missing PhCOO^- , SO_4^{2-} and Cl^- and the application of temperature stimulation, while the transformation from Ag43 to Ag14 does not require the addition of additional reactants and only temperature stimulation.

Reviewer #3 (Remarks to the Author):

This manuscript reports the stabilization of two novel Ag nanoclusters by metalloligands resulting from the combination of a thiacalix[4]arene and polyoxovanadates. Such a combination is unprecedented and the resulting structures are actually particularly original. Furthermore, owing to the probable variability in the way the thiacalixarene and the polyoxovanadate proligands can assemble, this work opens the door for a likely rich and innovative chemistry of nanoclusters protected by metalloligands. This is all the more interesting as the reported compounds show exceptional performances in photothermal conversion. This outstanding work deserves publication in Nature Comm., providing minor modifications as detailed below.

Response: Thanks for these positive comments on the novelty and significance of our study. We also believe that the revised manuscript improved quality thanks to your comments. In the following, we provide responses to the comments and suggestions point-by-point.

1) The Ag coordination spheres would deserve some description.

Response: Thank you for your constructive suggestion. In Ag14, 12 silver atoms at the waist are coordinated to TC4A⁴⁻-POVs metalloligands and CyS⁻ ligands, 2 silver atoms at the upper and lower ends of Ag14 are coordinated to three thioether groups of three TC4A⁴⁻, with the silver atoms forming the coordination numbers of 3, 4 and 5. For Ag43, all the silver atoms are coordinated with [(TC4A)₆(V₉O₁₆)]¹¹⁻ metalloligands, in addition, CyS⁻, PhCOO⁻, SO₄²⁻, S²⁻, Cl⁻ and DMF are involved in the coordination of silver atoms, with the silver atoms forming the coordination numbers of 3, 4 and 5.

In the revised version, we added these comments on the coordination between silver atom and S, Cl and/or O into the structure description section of the main text: "The coordination numbers of silver atoms in Ag14 are three (1 Ag atom in AgS₂O and 2 in AgS₃), four (4 in AgS₂O₂) and five (7 in AgS₂O₃)." and "Ag43 is protected by metalloligands, organic and inorganic mixed ligand shell, with the silver atoms forming the coordination numbers of three (3 Ag atoms in AgS₂O and 3 in AgSO₂), four (3 in AgS₂O₂, 3 in AgSO₂Cl and 3 in AgS₂OCl) and five (9 in AgS₂O₃, 6 in AgSO₃Cl, 3 in

AgS₂O₂Cl, 1 in AgSO₃Cl, 6 in AgSO₄ and 3 in AgS₂O₃).".

2) Is there any rational explanation for the outstanding (but somewhat unexpected) photothermal properties of compound Ag14?

Response: Thanks for your constructive suggestion. The main method of energy transition in photophysical process is illustrated with Jablonski diagram, containing the following several processes: excitation (or absorption), vibrational relaxation (heat), radiative emission (fluorescence) and non-radiative transition (heat), which clearly show the mechanism of energy release after excitation of specific molecules (Figure R2, same as Supplementary Fig. 16). All processes are competitive except for excitation, so superior photothermal materials need to have strong absorbance at specific wavelength as well as low fluorescence quantum yields. Both Ag14 and Ag43 have a strong absorbance at 660 nm in the solid state and do not have fluorescence, so the main energy release pathway is non-radiative transition (heat). As shown in UV-Vis absorption spectra (Supplementary Fig. 15a), we can clearly see that Ag14 in the solid state has a stronger absorption than Ag43 at 660 nm. Therefore, we attribute the better photothermal performance of Ag14 than Ag43 to the stronger absorption at 660 nm. Accordingly, we have interpreted the outstanding photothermal property of Ag14 and added the following discussion into the photothermal conversion section of the main text: "The energy transition in photophysical process is mainly illustrated by Jablonski diagram,⁴² which contains the following several processes: excitation (or absorption), vibrational relaxation (heat), radiative emission (fluorescence) and non-radiative transition (heat) (Supplementary Fig. 16).^{43,44}" and "As we have mentioned above, the channels of radiative decay of two NCs are almost suppressed. It is speculated that absorption may be the main factor influencing the photothermal conversion performance, and Ag14 has stronger absorption at 660 nm than Ag43 (Supplementary Fig. 15a), therefore, Ag14 can harvest more energy and produce higher temperature under 660 nm laser irradiation.".

Figure R2. Simplified Jablonski diagram illustrating photophysical processes of Ag14 and Ag43 at room temperature. PL: photoluminescence, VR: vibrational relaxation, NR: non-radiative process.

Furthermore, the related references were also listed in the "References" section of the main text:"

References:

42. Qi, J. et al. Light-driven transformable optical agent with adaptive functions for boosting cancer surgery outcomes. Nat. Commun. 9, 1848 (2018).

43. Pham, T.-T. D., Phan, L. M. T., Cho, S. & Park, J. Enhancement approaches for photothermal conversion of donor-acceptor conjugated polymer for photothermal therapy: a review. Sci. Technol. Adv. Mater. 23, 707-734 (2022).

44. Feng, G., Zhang, G. Q. & Ding, D. Design of superior phototheranostic agents guided by Jablonski diagrams. Chem. Soc. Rev. 49, 8179-8234 (2020)."

3) In these compounds, Ag is in the +I oxidation state. By using a reducing agent stronger than DMF, would it be possible to generate partly reduced $(Ag)_n$ nanoclusters, or perhaps vanadium(+IV) centers, with the possibility for electron transfer between metals?

Response: Thanks for your constructive suggestion. Using stronger reducing agents than DMF may facilitate the reduction of Ag(I) as well as inhibit the oxidation of vanadium(IV) to vanadium(V). Other stronger reducing agents available in our laboratory, such as Ph_2SiH_2 , $NaBH_3CN$, tBu_4NBH_4 , Me_4NBH_4 and

NaBH₄, were used in the synthesis experiments. Unfortunately, when the above reducing agents with stronger reducing ability than DMF were added, the reaction solution was prone to precipitation during stirring and neither produced desired clusters, presumably the Ag(I) might be directly reduced to silver nanoparticles.

Reviewers' Comments:

Reviewer #1:

Remarks to the Author:

I think the authors have well addressed my previous concerns, and I would like to recommend its acceptance for publication.

Reviewer #2:

Remarks to the Author:

The authors have satisfactorily responded to all of the reviewers' comments.
I can recommend acceptance to Nature Communications.

Reviewer #3:

Remarks to the Author:

In this revised version, the authors have satisfactorily taken into account the comments of the reviewers and improved their manuscript. This exceptional work can now be published as is.

RESPONSE TO REVIEWERS' COMMENTS:

Reviewer #1 (Remarks to the Author):

I think the authors have well addressed my previous concerns, and I would like to recommend its acceptance for publication.

Response: Thank you very much. We appreciate your recommendation for the publication of our work in Nature Communications.

Reviewer #2 (Remarks to the Author):

The authors have satisfactorily responded to all of the reviewers' comments.

I can recommend acceptance to Nature Communications.

Response: Thank you very much. We appreciate your recommendation for the publication of our work in Nature Communications.

Reviewer #3 (Remarks to the Author):

In this revised version, the authors have satisfactorily taken into account the comments of the reviewers and improved their manuscript. This exceptional work can now be published as is.

Response: Thank you very much. We appreciate the reviewer's for understanding our efforts to improve the paper and for recommending our work for publication in Nature Communications.